# Towards Effective GANs
# for Data Distributions with Diverse Modes

## Abstract

Generative Adversarial Networks (GANs), when trained on large datasets with diverse modes, are known to produce conflated images which do not distinctly belong to any of the modes. We hypothesize that this problem occurs due to the interaction between two facts: (1) For datasets with large variety, it is likely that the modes lie on separate manifolds. (2) The generator (G) is formulated as a continuous function, and the input noise is derived from a connected set, due to which G's output is a connected set. If G covers all modes, then there must be some portion of G's output which connects them. This corresponds to undesirable, conflated images. We develop theoretical arguments to support these intuitions. We propose a novel method to break the second assumption via learnable discontinuities in the latent noise space. Equivalently, it can be viewed as training several generators, thus creating discontinuities in the G function. We also augment the GAN formulation with a classifier C that predicts which noise partition/generator produced the output images, encouraging diversity between each partition/generator. We experiment on MNIST, celebA, STL-10, and a difficult dataset with clearly distinct modes, and show that the noise partitions correspond to different modes of the data distribution, and produce images of superior quality.

## 1 Introduction

Generative Adversarial Networks (Goodfellow et al., 2014) are powerful generative models that have enjoyed significant attention from the research community in the past few years. Despite several successes, the original formulation for GANs is widely acknowledged to be notoriously difficult to train due to instability issues. In particular, GANs face the mode collapse problem, where the generator resorts to generating a handful of samples which are assigned high probability by the discriminator. Several methods have been introduced to fix the mode collapse problem. (Che et al. (2016), Arjovsky et al. (2017), Gulrajani et al. (2017), Mao et al. (2016), Sun et al. (2017))

Despite improvements, state-of-art GANs still fail to generate meaningful samples on diverse and complex datasets such as ImageNet (Deng et al., 2009). GANs trained on such datasets produce conflated images which do not distinctly belong to any of the modes present in the dataset.

We hypothesize that this problem occurs due to the continuous nature of the generator function, along with the connectedness of the latent noise space, due to which the output set of the generator is also connected. This poses a problem when dealing with complex real life datasets with varied modes. Strong empirical and theoretical evidence suggests that real life images lie on low-dimensional manifolds (Narayanan & Mitter, 2010). It is highly probable that distinct modes (say bedroom images and human face images) lie on disjoint manifolds. If we assume that the generator does not suffer from the mode dropping problem, it must cover all these manifolds in its output. However, the output set being connected, must contain parts which do not belong to any of the manifolds, but simply join them.

We refer to such parts of the output as tunnels, since they connect otherwise disjoint manifolds. Tunnels do not resemble any of the images in the dataset, and are not similar to any of the modes. They correspond to the conflated images produced by the generator, and are undesirable. By this reasoning, we suggest that GANs with continuous generators and connected latent noise sets must suffer either from a certain degree of mode dropping or from producing conflated, garbled outputs when trained on complex and varied datasets like ImageNet.

We develop methods that allow GANs to cover disjoint manifolds without the use of tunnels, while not compromising on mode coverage. Our approach is to create learnable discontinuities in the latent noise space. This is done by learning $N$ different linear mappings (partitions) in the input layer of the generator. A noise vector (sampled from the standard normal distribution), gets mapped to $N$ different vectors by the input layer, and the rest of the processing remains the same as in standard generators. The output set of each mapping is a connected set, but the union of the $N$ output sets could potentially be disconnected. Thus, we break the connectedness assumption leading to the existence of tunnels. To facilitate learning distinct modes by each partition, we introduce a classifier that predicts which partition created a given input. We modify the loss functions to adjust for this change.

We experiment on standard datasets: MNIST (LeCun et al., 2010), celebA (Liu et al., 2015), STL-10 (a subset of ImageNet) (Coates et al., 2011), and a tough artificial dataset with very distinct modes - an equal mixture of LSUN (Yu et al., 2015) bedrooms and celebA, to verify the efficacy of our method. We compare our results with one of the best performing GAN variant (Gulrajani et al., 2017), and show an improvement in quality.

The major contributions of the paper are summarized below:

1. We identify a key problem with training GANs on large & diverse datasets, and provide intuition to explain its cause
2. We develop theoretical analyses to support and introduce rigor in the intuitions provided
3. Motivated by these analyses, we introduce a novel GAN setup to alleviate the problem
4. We experiment on a variety of standard datasets and report improvements over state-of-art formulations

## 2 RELATED WORK

Goodfellow et al. (2014) formulated GAN as a minimax game between two neural networks: generator $G_\theta$ and discriminator $D_\phi$. $G_\theta$ takes a random noise vector $\mathbf{z}$ as input and generates sample $G_\theta(\mathbf{z})$, while $D_\phi$ identifies whether input sample is real or generated by the generator $G_\theta$. Both $G_\theta$ and $D_\phi$ play a two-player minimax game with value function $V(G, D)$:

$$V(G, D) = \min_{\boldsymbol{\theta}} \max_{\boldsymbol{\phi}} \mathbb{E}_{\mathbf{x} \sim \mathbb{P}_r(\mathbf{x})}[log(D_\phi(\mathbf{x}))] + \mathbb{E}_{\mathbf{z} \sim \mathbb{P}(\mathbf{z})}[log(1 - D_\phi(G_\theta(\mathbf{z})))]$$

where $\mathbb{P}_r(\mathbf{x})$ is the real data distribution, and $\mathbb{P}(\mathbf{z})$ is arbitrary noise distribution (typically uniform or normal distribution). In practice, training GANs using above formulation is highly unstable and requires careful balance of generator and discriminator updates.

Radford et al. (2015) proposed a class of CNNs called DCGANs (Deep Convolutional GANs) with certain architectural specifications, and demonstrated better image quality than non-convolutional vanilla GAN architecture. Denton et al. (2015) used Laplacian pyramid framework for the generator, where a separate generative convnet model is trained using GAN approach at each level of pyramid, to generate images in coarse-to-fine fashion.

Despite better architectures, GANs suffered from problems like unstable training, vanishing gradients of generator, mode collapse. Salimans et al. (2016) proposed several heuristics such as feature matching, minibatch discrimination, historical averaging, label smoothing, primarily to stabilize GAN training.

Che et al. (2016) observed that GAN training can push probability mass in wrong direction, hence are prone to missing modes of data. They proposed regularization techniques to stabilize GAN training and alleviate mode missing problem by fair distribution of probability mass across modes of the real data distribution.

Arjovsky & Bottou (2017) provided theoretical analysis of training dynamics of GANs, and problems including instability and saturation. They revealed fundamental problems with original GAN formulation and provided directions towards solving them.

Several papers proposed alternative objective function of generator and discriminator. Arjovsky et al. (2017), Gulrajani et al. (2017) proposed new loss function which approximately minimizes

Wasserstein distance between real and generated data distribution instead of Jensen Shannon Divergence. They claim their formulation does not require careful balance between generator and discriminator updates, thus lead to stable training without saturating the gradients. They observed no evidence of mode collapse in their experiments. Mao et al. (2016) used squared-loss instead of log-loss in original formulation, which provides generator with better non-vanishing gradients. Zhao et al. (2016) view discriminator as an energy function making it possible to use additional loss functions other than logistic output binary classifier, which was found to stabilize GAN training. Sun et al. (2017) propose to train discriminator based on linear separability between hidden representation of real and generated samples and train generator based on decision hyperplanes between hidden representations computed using Linear Discriminant Analysis.

For labelled datasets, Mirza & Osindero (2014), Odena et al. (2016) employed label conditioning in both generator and discriminator to generate discriminable and diverse samples across classes. While this helps produce better samples for complex datasets, it requires the presence of labelled data. In this paper we propose methods to improve performance of GANs on complex datasets without making use of labels.

## 3    PROBLEMS WITH LEARNING DIVERSE DATASETS

In this section, we further develop the ideas from the introduction. We also provide theoretical analyses to lend support to these ideas.

### 3.1    TOPOLOGY OF DIVERSE DISTRIBUTIONS

Theoretical analyses and empirical studies suggest that probability distributions of real images (denoted by $\mathbb{P}_r$) have supports that lie on low dimensional manifolds (Arjovsky & Bottou (2017), Narayanan & Mitter (2010)).

We choose to represent the support set $S$ of distribution $\mathbb{P}_r$ by the set of its connected components, i.e. the set of maximal connected subsets of $S$. These components are disjoint and their union is $S$. In other words, they form a partition of $S$. As suggested earlier, each component is a low-dimensional manifold in high-dimensional space. Throughout this paper, we use the terms manifold of $S$ and connected component of $S$ interchangeably, unless mentioned otherwise.

Consider highly distinct images $m_1, m_2$ of a complex and varied dataset. These images cannot be path-connected, since that would imply the existence of a smooth sequence of images in $S$ starting from $m_1$, and slowly transitioning to $m_2$. Such a sequence clearly does not exist for very distinct $m_1, m_2$, e.g. in a dataset consisting of face and bedroom images, if $m_1$ is a face, and $m_2$ is a bedroom, it is not possible for a sequence of realistic facial and bedroom images to smoothly transition from $m_1$ to $m_2$. Since path-connectedness and connectedness are equivalent properties in open subsets of $\mathbb{R}^n$, the manifolds on which $m_1, m_2$ lie, must be disconnected, and hence must be separate components of $S$.

We summarize the above discussion with the following results:

**Result 1.** *Sufficiently distinct images from the support set of real-life image probability distributions must lie on disconnected manifolds.*

As a corollary, we have:

**Result 2.** *The support set of a real-life image probability distribution must have at least as many connected components as the size of the maximal set of (sufficiently, pairwise) distinct modes.*

### 3.2    TOPOLOGY OF GENERATOR'S OUTPUT SET

The generator is often formulated as a continuous function of the input latent noise vector. Moreover, the probability distribution from which the noise vector is drawn is usually a simple distribution like Gaussian or uniform, which have connected support sets.

Finally, we observe that a continuous function, applied to a connected set results in an output set which is connected. Thus we have:

**Result 3.** *The output set produced by a continuous generator acting on a latent noise distribution with connected support set must be connected.*

### 3.3 LESSER OF TWO EVILS

From the previous discussions, we know that the output of the generator must be connected, while diverse distributions must lie on supports consisting of several disconnected manifolds.

If the generator does not suffer from mode dropping, then it must cover all the distinct modes in its output set. However, this implies the existence of parts in the generator's output which connect these disconnected manifolds. We call such parts of the output space as tunnels. Since tunnels do not exist in the real distribution, they must correspond to unrealistic images.

We condense these ideas to the following result:

**Result 4.** *A continuous generator, with inputs drawn from a connected set, must either suffer from significant mode dropping or generate unrealistic images to some degree.*

In the rest of the discussion, we assume that mode dropping is not a significant problem. This is a realistic assumption due to several heuristics and formulations that alleviate this problem (Salimans et al. (2016), Che et al. (2016), Arjovsky et al. (2017), Gulrajani et al. (2017), Sun et al. (2017)). We concentrate on the problem of generation of unrealistic, distorted outputs.

### 3.4 UNUSED MEASURE IN LIPSCHITZ CONTINUOUS GENERATORS

If we assume that the generator is $K-$Lipschitz (this happens with a variety of regularization terms added to the generator's loss), i.e. for any two points $z_1, z_2$ in the latent noise space we have

$$\frac{\|G(z_1) - G(z_2)\|}{\|z_1 - z_2\|} \leq K$$

then the generator's output must gradually shift from one manifold to another as we travel in $Z$ space, since the slope is bounded.

The region of shift does not belong to any outputs in the real distribution. It is simple to see that some measure of the probability distribution $Z$ is wasted on mapping these unwanted outputs.

To demonstrate this, we consider the simple case where $\mathbb{P}_r$ consists of equal measure of samples of type A and type B, both of which are disconnected and highly distinct from each other. For simplicity, we assume that the latent noise $z$ is drawn from a 1-D region $[-1, 1]$, with uniform probability.

Let the distance between set A and set B (defined as $\inf_{a \in A, b \in B} \|a - b\|$) be $\beta$ ($> 0$ due to high distinction between the sets). Let $\mathcal{M}_A, \mathcal{M}_B$ be subsets of $Z$ mapping to $A$ and $B$ respectively.

For any arbitrary $z_1 \in \mathcal{M}_A$ and $z_2 \in \mathcal{M}_B$, from the Lipschitz condition, we have:

$$K \geq \frac{\|G(z_1) - G(z_2)\|}{|z_1 - z_2|} \geq \frac{\beta}{|z_1 - z_2|}$$

$$\implies |z_1 - z_2| \geq \frac{\beta}{K}$$

Hence, the distance between any two points of $\mathcal{M}_A$ and $\mathcal{M}_B$ must be at least $\frac{\beta}{K}$, as a result of which, the distance between the sets $\mathcal{M}_A$ and $\mathcal{M}_B$ is at least $\frac{\beta}{K}$.

Clearly, in our hypothetical case, this results in a gap of at least $\frac{\beta}{K}$ wherever $\mathcal{M}_A$ ends and $\mathcal{M}_B$ begins. Since we have assumed uniform distribution, a probability measure of $\frac{\beta}{2K}$ is lost to undesired outputs.

### 3.5 CONDITIONAL GANS WORK BETTER ON DIVERSE DATASETS

It is well-known that conditional GANs (Mirza & Osindero (2014), Odena et al. (2016)) are better at learning complex datasets. The label information incorporated in the inputs of the generator plays a large role in making better outputs.

However, we also believe that the discontinuity introduced by the one-hot label representations contributes significantly to improving performance. More concretely, the input to the generator is a latent noise vector, along with the one-hot label representation. Hence, the input space is partitioned into $n$ (number of classes) disconnected components due to the discrete nature of the one-hot labels.

This breaks the connectedness assumption of the input space, which was a crucial part of the problem's cause. Note however that conditional GANs require labeled inputs, which may not be available for many datasets.

## 4    PROPOSED SOLUTIONS

Central to the results in section 3 were the assumptions of continuity of the generator and the connectedness of the support set of the latent noise distribution. Breaking either of these assumptions could lead to potential solutions. We now describe novel GAN formulations that breaks these assumptions while not requiring any labeled data.

### 4.1    MODEL

We create trainable discontinuities in the latent noise space by learning $N$ different linear mappings $\{L_1, L_2 \ldots L_N\}$ in the input layer of the generator.

A noise vector $z$ gets mapped to $N$ vectors $\{y_1 = L_1(z) \ldots y_N = L_N(z)\}$ by the input layer, and the rest of the processing remains the same as in standard generators. We end with $N$ outputs $\{g_1 = G(y_1), g_2 = G(y_2) \ldots g_N = G(y_N)\}$.

Each of the linear layers $L_i$ maps the input latent noise space $Z$ to a connected output set $O_i$. While $O_i$ is a connected set, the union of the $N$ output sets $(O_1 \ldots O_N)$ could potentially be disconnected (or each mapping could collapse to the same matrix, if required by the data).

This approach of partitioning the noise space can also be seen as training $N$ different generators, with shared parameters, except for the input layer. In this view, the generator function has become potentially discontinuous due to indexability of the outputs (i.e. choosing which output to take).

In either view, we have broken one of the assumptions leading to the existence of tunnels.

Finally, to facilitate the learning of distinct modes by each partition (or generator), we introduce a classifier $C$ that predicts which partition (or generator) created a given input. We modify the GAN value function to suitably account for this change.

We would like to emphasize that this formulation is generic, and can be plugged in with different types of generators, discriminators, partition mappings, and classifiers. Moreover, any improvements in GAN formulations can be independently incorporated into our setup. However, we do make specific design choices for the purpose of this paper, which we describe below.

To reduce the cost of training, we take $C$ to be a linear mapping operating on the last hidden layer of the discriminator. We believe that the discriminator extracts useful abstract features for judging the samples produced by the generator, and these can be effectively reused for partition classification.

### 4.2    MODIFIED LOSS FUNCTIONS

We add the partition classification loss to the existing generator and discriminator losses from the chosen base formulation. The new loss functions are:

$$\hat{L}_D = L_D + \alpha \mathbb{E}_{\mathbf{x} \sim \mathbb{P}_g}[L_c(y, C(\mathbf{x}))]$$
$$\hat{L}_G = L_G + \alpha \mathbb{E}_{\mathbf{x} \sim \mathbb{P}_g}[L_c(y, C(\mathbf{x}))]$$

$L_D, L_G$ are the original loss functions for the discriminator and generator respectively, in the base GAN formulation. For vanilla GANs,

$$L_D = \mathbb{E}_{\mathbf{x} \sim \mathbb{P}_r}[\log D(\mathbf{x})] - \mathbb{E}_{\mathbf{x} \sim \mathbb{P}_g}[\log(1 - D(\mathbf{x}))]$$
$$L_G = \mathbb{E}_{\mathbf{x} \sim \mathbb{P}_g}[\log(1 - D(\mathbf{x}))]$$

Here $L_c(y, C(x))$ is the cross-entropy classification loss for input image $\mathbf{x}$, which was generated by partition $y$, and $C(\mathbf{x})$ is the classifier's softmax output vector.

$\alpha$ is a hyperparameter introduced to control the relative importance of generating good samples w.r.t encouraging diversity between the outputs of different partitions.

Finally, we must describe the exact meaning of $\mathbb{P}_g$ in our formulation, since there are $N$ generators. We sample the generator at each training step uniformly, thus making $\mathbb{P}_g = \frac{1}{N} \sum_{i=1}^{N} \mathbb{P}_{g_i}$, where $\mathbb{P}_{g_i}$ is the probability distribution induced by generator $i$.

### 4.3 MIXTURE-OF-GAUSSIANS LATENT DISTRIBUTION

We now propose a simpler method for implementing a softer version of disconnectedness for the latent noise distribution.

In this method, the noise vectors are drawn from a mixture-of-Gaussians with trainable means $(\boldsymbol{\mu}_1, \ldots, \boldsymbol{\mu}_N)$ and covariance matrices $(Diag(\boldsymbol{\sigma}_1), \ldots, Diag(\boldsymbol{\sigma}_N))$. If required by the data, the means can move sufficiently far away from each other during training, leading to an almost disconnected distribution. Each Gaussian is given equal probability weight, i.e., it is chosen uniformly at random.

Our implementation makes use of the reparameterization trick (Kingma & Welling (2013), Doersch (2016)). We sample $\mathbf{z} \sim \mathcal{N}(\mathbf{0}, \mathbf{I})$, along with an index $i \sim Uniform(\{1, \ldots, N\})$. We then use $\hat{\mathbf{z}} = \mathbf{z} \odot \boldsymbol{\sigma}_i + \boldsymbol{\mu}_i$ as the latent vector for generating the sample. We let the gradients backpropagate to $\boldsymbol{\mu}_i$ and $\boldsymbol{\sigma}_i$, allowing for a learnable latent distribution.

Additionally, as mentioned before, we may use a classifier to predict the Gaussian that generated a given sample, hence encouraging diversity between different Gaussians.

## 5 EXPERIMENTAL SETUP

### 5.1 DATASETS

We experiment with the image generation task on MNIST, celebA consisting of facial images, STL-10, and an artificial dataset called CelebRoom consisting of $100,000$ images randomly sampled from celebA and $100,000$ images randomly sampled from LSUN bedroom dataset. CelebRoom was constructed with the explicit goal of including diverse modes, making it difficult for GANs to train.

As in Gulrajani et al. (2017), we present results on a toy dataset sampled from an equal mixture 8 bivariate Gaussians. The means are arranged uniformly in a circle with radius 2, and the covariance matrices are set to $0.02\mathbf{I}$.

### 5.2 IMPLEMENTATION

We modified a popular TensorFlow (Abadi et al., 2015) implementation of the WGAN-GP architecture[1] for all experiments.

### 5.3 HYPERPARAMETER TUNING

$N$ is fixed for each dataset. We use $N = 10$ for MNIST, and $N = 8$ for celebA, STL-10, and CelebRoom.

$\alpha$ is fixed to $1.0$ for all experiments with multiple partitions.

We use the ADAM optimizer (Kingma & Ba, 2014) for all experiments.

For each dataset (except MNIST), we compare the samples obtained using the following setups:

For MNIST and celebRoom, we also present results generated from the mixture-of-Gaussians latent distribution method.

---

[1] https://github.com/igul222/improved_wgan_training

Table 1: Experimental Setups

| Architecture | Base Loss Formulation | Multiple Partitions |
|---|---|---|
| DCGAN | Vanilla | No |
| ResNet | Vanilla | No |
| DCGAN | WGAN-GP | No |
| ResNet | WGAN-GP | No |
| DCGAN | Vanilla | Yes |
| ResNet | Vanilla | Yes |
| DCGAN | WGAN-GP | Yes |
| DCGAN | WGAN-GP | Yes |

We used the hyperparameters suggested by the original papers for each architecture. The ResNet (He et al., 2016) architecture used has 4 residual blocks, as described in Gulrajani et al. (2017).

We avoid intensive hyperparameter tuning in order to encourage discovery of methods with relatively high hyperparameter insensitivity.

We also tried training multiple partitions with the Least Squares GAN (Mao et al., 2016) formulation, but noticed that the training quickly diverged. Hence we do not report these results here.

## 5.4 SAMPLES

We present samples on the five datasets. For samples generated using multiple partitions, each column represents the images produced by a single partition.

## 6 RESULTS

### 6.1 8 GAUSSIANS TOY DATASET

We observe that Vanilla GAN is unable to cover all Gaussians. As training proceeds, it shuttles between covering different subsets of Gaussians.

WGAN-GP takes a long time to converge. It is finally able to cover all Gaussians, but not satisfactorily, with several samples lying between different means. We note that by reducing the gradient penalty significantly, we are able to achieve faster convergence. However, we chose to use the recommended hyperparameters for all formulations.

Vanilla GAN with partitions is able to cover all the modes, and converges quickly.

### 6.2 MNIST

We observe a slight improvement in quality in the samples generated by our model when using the partition classifier. The images produced with the classifier formulation are more distinct, sharper, and do not contain spurious appendages.

We expected each partition to produce a single type of digit, but this does not seem to be the case. Despite this, we do note that the partition classifier's loss was very close to zero upon successful training. This signifies that the partitions did learn to map distinct types of outputs, but not by digit label.

This can be explained by the fact that labels are not the only way to classify digits. There are several combinations of digit style, tilt, width etc. that could be captured by the partitions.

### 6.3 CELEBA

We observed slight improvements in quality in the samples generated by our model over the WGAN + Gradient Penalty baseline with the DCGAN architecture.

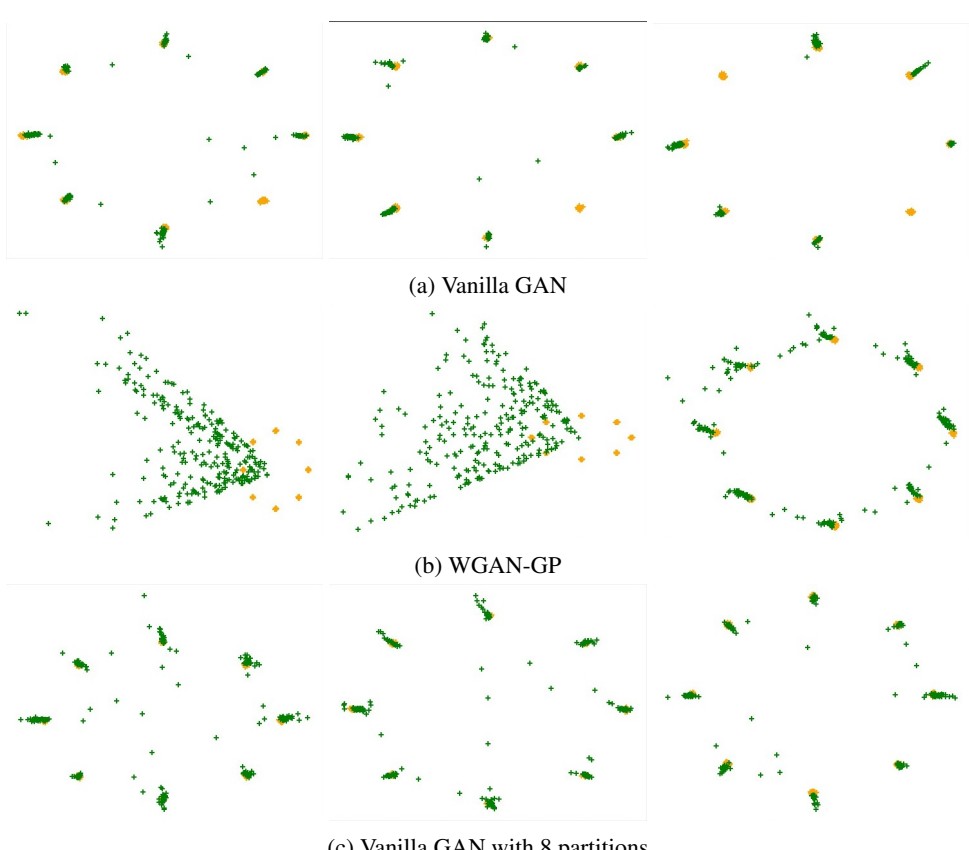

(a) Vanilla GAN

(b) WGAN-GP

(c) Vanilla GAN with 8 partitions

Figure 1: Samples generated by Vanilla GAN, WGAN-GP, Vanilla GAN with multiple partitions on a toy dataset with 8 Gaussians. Yellow points represent true samples, while green points represent samples generated by GAN. The first image, second image, and third image in each row show samples after 10000, 15000, and 20000 training steps. All 3 models use an MLP with 3 hidden layers with 512 nodes.

Figure 2: $28 \times 28$ MNIST samples generated using our method: without generator prediction (left), with generator prediction (centre), with mixture-of-Gaussians latent distribution along with Gaussian prediction (right)

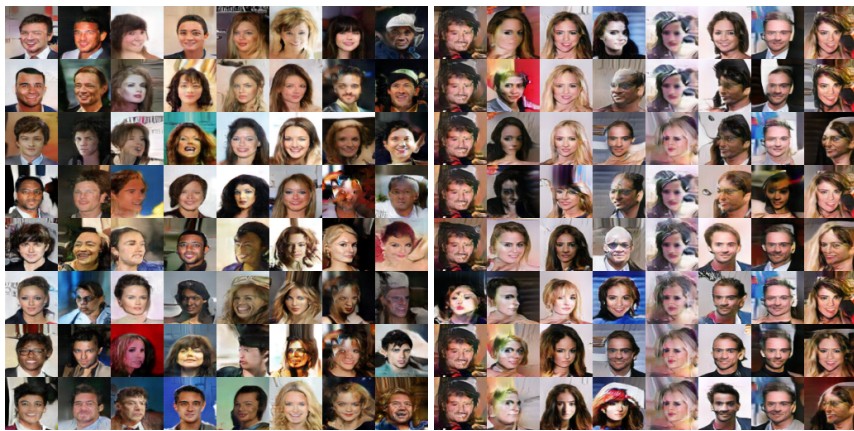

Figure 3: $64 \times 64$ celebA samples generated using vanilla loss with 8 partitions and partition prediction using: DCGAN architecture (left), ResNet architecture (right)

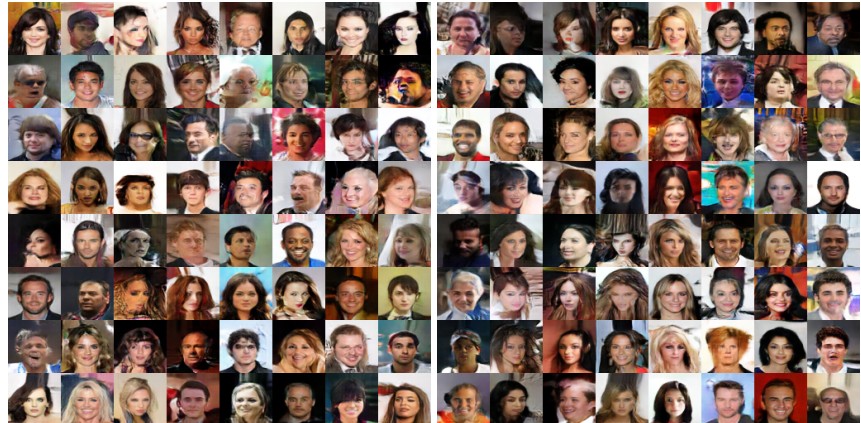

Figure 4: $64 \times 64$ celebA samples generated using DCGAN architecture and WGAN-GP loss with: no partitions (left), 8 partitions with partition prediction (right)

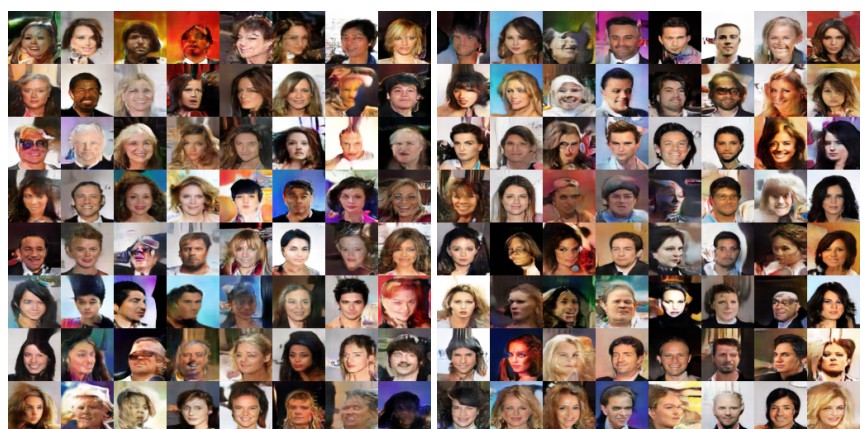

Figure 5: $64 \times 64$ celebA samples generated using ResNet architecture and WGAN-GP loss with: no partitions (left), 8 partitions with partition prediction (right)

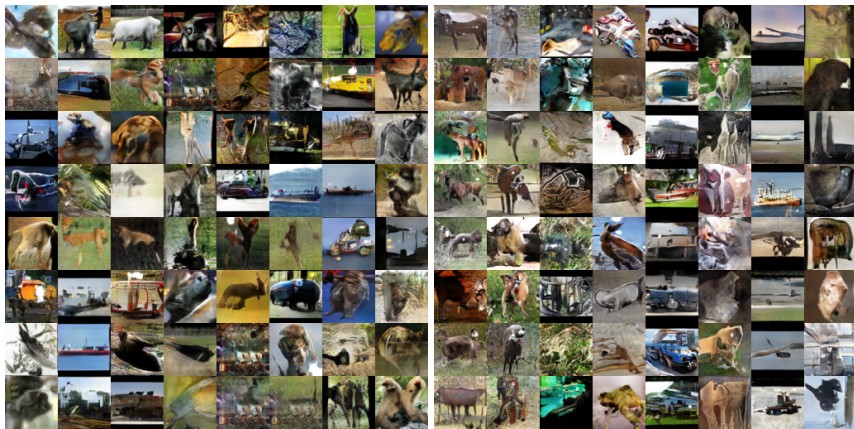

Figure 6: $64 \times 64$ STL-10 samples generated using DCGAN architecture with vanilla loss with: No partitions (left), 8 partitions with partition prediction (right)

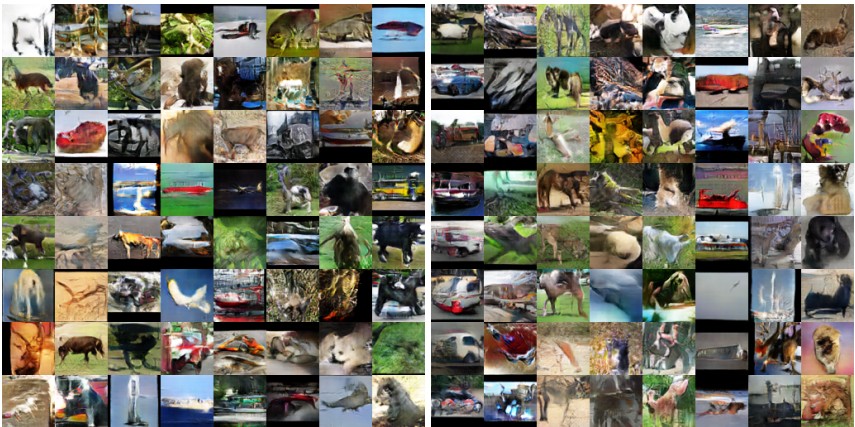

Figure 7: $64 \times 64$ STL-10 samples generated using DCGAN architecture with WGAN-GP loss with: No partitions (left), 8 partitions with partition prediction (right)

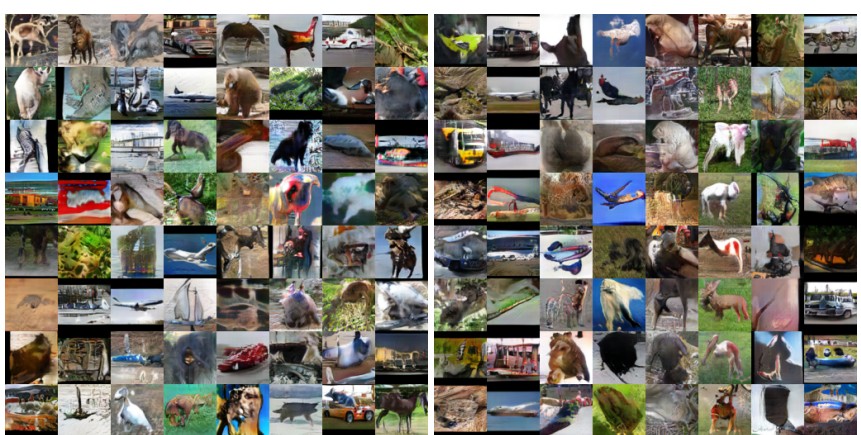

Figure 8: $64 \times 64$ STL-10 samples generated using ResNet architecture with WGAN-GP loss with: No partitions (left), 8 partitions with partition prediction (right)

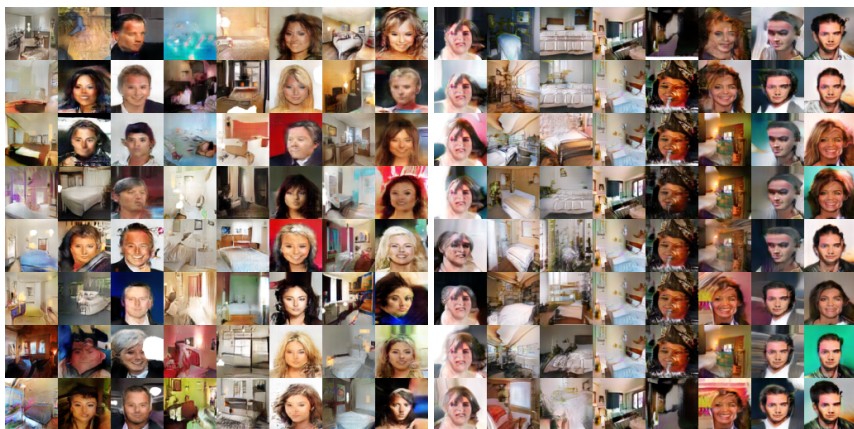

Figure 9: $64 \times 64$ celebRoom samples generated using vanilla loss with 8 partitions and partition prediction with: DCGAN architecture (left), ResNet architecture (right)

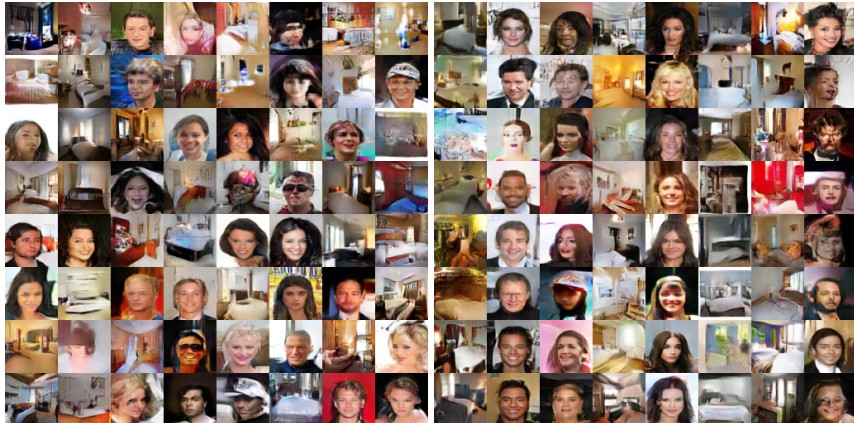

Figure 10: $64 \times 64$ celebRoom samples generated using DCGAN architecture and WGAN-GP loss with: No partitions (left), 8 partitions with partition prediction (right)

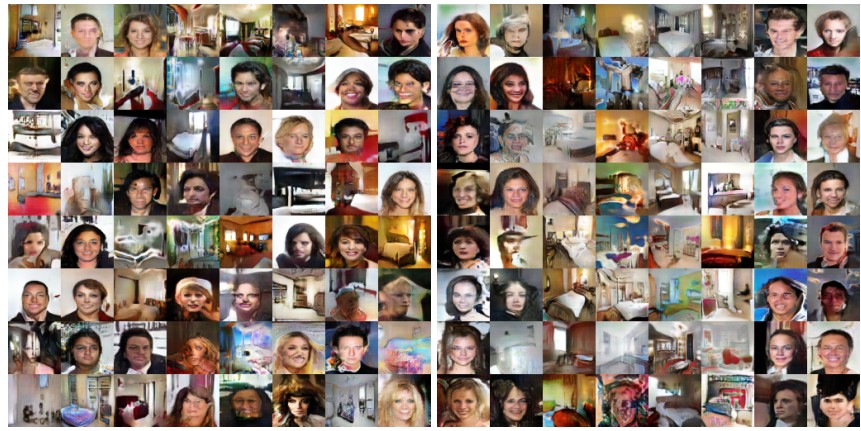

Figure 11: $64 \times 64$ celebRoom samples generated using ResNet architecture and WGAN-GP loss with: No partitions (left), 8 partitions with partition prediction (right)

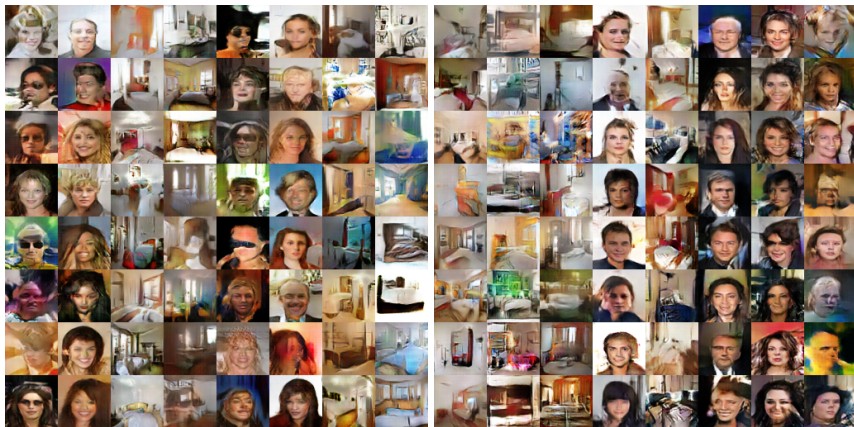

Figure 12: $64 \times 64$ celebRoom samples generated using DCGAN architecture with mixture-of-Gaussians latent distribution using: no prediction of Gaussians (left), prediction of Gaussians (right)

Each partition seems to produce similar types of images, both with and without a partition classifier formulation to encourage diversity. For instance, the fifth partition (fifth column) in DCGAN architecture and WGAN-GP loss seems to capture the mode of smiling women, while the eigth partition (last column) seems to generate faces of men (often bespectacled).

We also note that the ResNet architecture is unable to train successfully with partitions. We notice heavy mode collapse, and noisy outputs.

### 6.4    STL-10

Our experiments show heavy conflation of modes in the outputs generated by all GAN formulations. However, this problem is ameliorated to some extent with our partition GAN.

In particular, we notice some partitions capturing distinctly well-formed modes. The DCGAN architecture with WGAN-GP loss with 8 partitions shows several such examples. The first partition (first column) seems to create images of vehicles, while the sixth partition (sixth column) seems to generate images of oceans and open skies (along with boats, ships, and airplanes). Similar clustering can also be observed with the ResNet architecture with WGAN-GP loss, where partition four (column four) generates images of birds and planes, while the sixth partition (column six) generates images of animals.

### 6.5    CELEBROOM

We observe significant conflation of bedrooms and facial images in this dataset with the baseline model. Our model alleviates this problem to some extent, but does not solve it completely. However, we see that each partition does either primarily create faces or bedrooms.

We would like to note that if a particular partition generates bad quality samples, it might be due to the inherent difficulty in creating that portion of the real distribution. In such cases, we can drop the outputs from that partition at the cost of not capturing $\frac{1}{N}$ of the distribution.

## 7    CONCLUSION

We highlighted a major problem in training GANs on complex image datasets and introduced theoretical analysis for the problem of generation of unrealistic, conflated images in such cases. We proposed the addition of discontinuity in latent noise space of the generator for covering disjoint and diverse modes of the data distribution, and augmented the loss functions to encourage diversity. We showed improvements over existing models without much hyperparameter tuning.

In future, we hope to perform an extensive exploration of the search space to obtain a set of hyper-parameters along with better methods to introduce discontinuities in the generator that perform well on a variety of datasets, while significantly improving image quality.

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
