# OpenReview forum: "Towards Effective GANs for Data Distributions with Diverse Modes"
_ICLR.cc/2018/Conference — Invite to Workshop Track_

### Official Review · AnonReviewer2 · 2017-11-27
**Important details are missing**

**Rating:** 4
**Confidence:** 3

**Review:**

The authors propose to train multiple generators (with same set of parameters), each of which with a different linear mapping in the first layer. The idea is that the final output of the generator should be a distribution whose support are disconnected. The idea does look interesting. But a lot of details is missing and needs clarification.

1) A lot of technical details are missing. The main formula is given in page 6 (Sec. 4.2), without much explanation. It is also not clear how different generators are combined as a final generator to feed into the discriminator. Also how are the diversity enforced?

2) The experiments are not convincing. It is stated that the new method produces results that are visually better than existing ones. But there is no evidence that this is actually due to the proposed idea. I would have liked to see some demonstration of how the different modes look like, how they are disconnected and collaborate to form a stronger generator. Even some synthetic examples could be helpful.

---

> ### Author Response · Authors · 2017-12-31
> **Reply: Important details are missing**
>
> We thank the reviewer for their insightful critique and detailed comments.
> We have added a revision of the paper with additional experiments, minor corrections & clarifications. We realize that there was an error in our discussion concerning unrealistic outputs in DCGANs, and we have withdrawn that section from the paper. However, we would like to point out that this does not detract our main message because this particular proof was meant to mathematically elucidate the problem of tunneling in DCGANs as an example. While our attempt to showcase the problem particularly for DCGANs stands invalidated, the rest of the general arguments set forth in the paper still hold.
> We address other pending concerns below:
>
> 1) A lot of technical details are missing. The main formula is given in page 6 (Sec. 4.2), without much explanation. It is also not clear how different generators are combined as a final generator to feed into the discriminator. Also how are the diversity enforced?
>
> A: We apologize for any obfuscation in our presentation, however, we do explain the setup in the Proposed Solutions section. We have rewritten the main formula to make it more understandable. We also address the specific queries raised here:
> i) The generators are "combined" by sampling uniformly from each of them. The resulting distribution is reported in the last paragraph of the Modified Loss Functions section.
> ii) The diversity is enforced by adding a prediction loss to the discriminator's and generator's losses. Thus, each generator is incentivized to produce outputs which are distinguishable from the outputs of the other generators. The discriminator is incentivized to learn features which help in distinguishing between the different generators.
>
> 2) The experiments are not convincing. It is stated that the new method produces results that are visually better than existing ones. But there is no evidence that this is actually due to the proposed idea. I would have liked to see some demonstration of how the different modes look like, how they are disconnected and collaborate to form a stronger generator. Even some synthetic examples could be helpful.
>
> A: Visualization (in fact even distinguishing) modes in high dimensional data is very hard, hence it is difficult to show how the partitions collaborate. We thank the reviewer for suggesting synthetic examples. We have included experiments from a popular toy setup consisting of 8 bivariate Gaussians arranged in a circle (thus 8 modes are present). We report results by running vanilla GAN, WGAN-GP, and our setup on this dataset, showing that the partitions can collaborate to cover distinct modes, which cannot be done with other setups. We have also included additional experiments on STL-10 (which is an ImageNet subset) as evidence for the efficacy of the split generator setup.

---

### Official Review · AnonReviewer3 · 2017-11-27
**Review for "Towards Effective GANs for Data Distributions with Diverse Modes"**

**Rating:** 6
**Confidence:** 5

**Review:**

Summary:

The paper studies the problem of learning distributions with disconnected support. The paper is very well written, and the analysis is mostly correct, with some important exceptions. However, there are a number of claims that are unverified, and very important baselines are missing. I suggest improving the paper taking into account the following remarks and I will strongly consider improving the score.

Detailed comments:

- The paper is very well written, which is a big plus.

- There are a number of claims in the paper that are not supported by experiments, citations, or a theorem.

- Sections 3.1 - 3.3 can be summarized to "Connected prior + continuous generator => connected support". Thus, to allow for disconnected support, the authors propose to have a discontinuous generator. However to me it seems that a trivial and important attack to this problem is to allow a simple disconnected prior, such as a mixture between uniforms, or at least an approximately disconnected (given the superexponential decay of the gaussian pdf) of a mixture of gaussians, which is very common. The authors fail to mention this obvious alternative, or explore it further, which I think weakens the paper.

- Another standard approach to attacking diverse datasets such as imagenet is adding noise in the intermediate layers of the generator (this was done by EBGAN and the Improved GAN paper by Salimans et al.). It seems to me that this baseline is missing.

- Section 3.4, paragraph 3, "the outputs corresponding to vectors linearly interpolated from z_1 to z_2 show a smooth". Actually, this is known to not perform very well often, indeed the interpolations are done through great circles in z_1 and z_2. See https://www.youtube.com/watch?v=myGAju4L7O8 for example.

- Lemma 1 is correct, but the analysis on the paragraph following is flat out wrong. The fact that a certain z has high density doesn't imply that the sample g_\theta(z) has high density! You're missing the Jacobian term appearing in the change of variables. Indeed, it's common to see neural nets spreading appart regions of high probability to the extent that each individual output point has low density (this is due in its totallity to the fact that ||\nabla_x g_\theta(z)|| can be big.

- Borrowing from the previous comment, the evidence to support result 5 is insufficient. I think the authors have the right intuition, but no evidence or citation is presented to motivate result 5. Indeed, DCGANs are known to have extremely sharp interpolations, suggesting that small jumps in z lead to large jumps in images, thus having the potential to assign low probability to tunnels.

- A citation, experiment or a theorem is missing showing that the K of a generator is small enough in an experiment with separated manifolds. Until that evidence is presented, section 3.5 is anecdotal.

- The second paragraph of section 3.6 is a very astute observation, but again it is necessary to show some evidence to verify this intuition.

- The authors then propose to partition the prior space by training separate first layers for the generator in a maximally discriminative way, and then at inference time just sampling which layer to use uniformly. It's important to note that this has a problem when the underlying separated manifolds in the data are not equiprobable. For example, if we use N = 2 in CelebRoom but we use 30% faces and 70% bedrooms, I would still expect tunneling due to the fact that one of the linear layers has to cover both faces and bedrooms.

- MNIST is known to be a very poor benchmark for image generation, and it should be avoided.

- I fail to see an improvement in quality in CelebA. It's nice to see some minor form on clustering when using generator's prediction, but this has been seen in many other algorithms (e.g. ALI) with much better results long before. I have to say also the official baseline for 64x64 images in wgangp (that I've used several times) gives much better results than the ones presented in this paper https://github.com/igul222/improved_wgan_training/blob/master/gan_64x64.py .

- The experiments in celebRoom are quite nice, and a good result, but we are still missing a detailed analysis for most of the assumptions and improvements claimed in the paper. It's very hard to make very precise claims about the improvements of this algorithm in such a complex setting without having even studied the standard baselines (e.g. noise at every layer of the generator, which has very public and well established code https://github.com/openai/improved-gan/blob/master/imagenet/generator.py).

- I would like to point a lot of tunneling issues can be seen and studied in toy datasets. The authors may want to consider doing targeted experiments to evaluate their assumptions.

=====================

After the rebuttal I've increased my score. The authors did a great job at addressing some of the concerns. I still think there is more room to be done as to justifying the approach, dealing properly with tunneling when we're not in the somewhat artificial case of equiprobable partitions, and primarily at understanding the extent to which tunneling is a problem in current methods. The revision is a step forward in this direction, but still a lot remains to be done. I would like to see simple targeted experiments aimed at testing how much and in what way tunneling is a problem in current methods before I see high dimensional non quantitative experiments.

In the case where the paper gets rejected I would highly recommend the acceptance at the workshop due to the paper raising interesting questions and hinting to a partial solution, even though the paper may not be at a state to be published at a conference venue like ICLR.

---

> ### Author Response · Authors · 2017-12-31
> **Reply: Review for "Towards Effective GANs for Data Distributions with Diverse Modes"**
>
> We thank the reviewer for their insightful critique and detailed comments.
> We have added a revision of the paper with additional experiments, minor corrections & clarifications. We realize that there was an error in our discussion concerning unrealistic outputs in DCGANs, and we have withdrawn that section from the paper. However, we would like to point out that this does not detract our main message because this particular proof was meant to mathematically elucidate the problem of tunneling in DCGANs as an example. While our attempt to showcase the problem particularly for DCGANs stands invalidated, the rest of the general arguments set forth in the paper still hold.
> We address other pending concerns below:
>
> Q: There are a number of claims in the paper that are not supported by experiments, citations, or a theorem.
>
> A: We shall do our best to provide any missing citations. We shall be grateful to the reviewer for directing us towards any specific unsupported claims.
>
> Q: ... it seems that a trivial and important attack to this problem is to allow a simple disconnected prior, such as a mixture between uniforms, or at least an approximately disconnected (given the superexponential decay of the gaussian pdf) of a mixture of gaussians, which is very common.
>
> A: We thank the reviewer for pointing out this omission. We did consider this alternative originally, in the form of a mixture of Gaussians with trainable parameters, but did not report it. We have included it in the revision, along with supporting experiments.
>
> Q: Lemma 1 is correct, but the analysis on the paragraph following is flat out wrong. The fact that a certain z has high density doesn't imply that the sample g_\theta(z) has high density! You're missing the Jacobian term appearing in the change of variables ... Borrowing from the previous comment, the evidence to support result 5 is insufficient ... Section 3.4, paragraph 3, "the outputs corresponding to vectors linearly interpolated from z_1 to z_2 show a smooth". Actually, this is known to not perform very well often, indeed the interpolations are done through great circles in z_1 and z_2.
>
> A: We thank the reviewer for spotting this error. Our attempt was to showcase the tunneling problem specifically for an easily understood example. However, in light of the technical error, we have withdrawn this discussion from the revised submission.
>
> Q: A citation, experiment or a theorem is missing showing that the K of a generator is small enough in an experiment with separated manifolds. Until that evidence is presented, section 3.5 is anecdotal.
>
> A: We agree that there is little support for the fact that K is small enough. However, we do not claim that K is indeed small. We just propose that there is some measure of probability lost, as a function of K.
>
> Q: The second paragraph of section 3.6 is a very astute observation, but again it is necessary to show some evidence to verify this intuition.
>
> A: We believe that one way to verify this observation would be to discard label information, while holding on to the partitioning property endowed by conditioning on labels. This is precisely what our experiments with multi-partition GANs do. We request the reviewer to consider the experiments in this light.
>
> Q: The authors then propose to partition the prior space by training separate first layers for the generator in a maximally discriminative way, and then at inference time just sampling which layer to use uniformly. It's important to note that this has a problem when the underlying separated manifolds in the data are not equiprobable. For example, if we use N = 2 in CelebRoom but we use 30% faces and 70% bedrooms, I would still expect tunneling due to the fact that one of the linear layers has to cover both faces and bedrooms.
>
> A: We agree with the reviewer that tunneling will still occur. However, it does get reduced to some extent by our method, since one entire generator can be devoted to creating 50% bedrooms. The other generator can create 30% faces and 20% bedrooms. Thus only this part will face tunneling issues, and the first generator escapes these issues.
>
> Q: I have to say also the official baseline for 64x64 images in wgangp (that I've used several times) gives much better results than the ones presented in this paper
>
> A: We thank the reviewer for pointing out the discrepancy, and directing us to the official code. As a precaution, we reimplemented our setup using the WGAN-GP code and reconducted all experiments.
>
> Q: I would like to point a lot of tunneling issues can be seen and studied in toy datasets. The authors may want to consider doing targeted experiments to evaluate their assumptions.
>
> A: We have included experiments from a popular toy setup consisting of 8 bivariate Gaussians arranged in a circle.
> We have also included results on STL-10, a subset of ImageNet as a step towards ImageNet complexity.

---

### Official Review · AnonReviewer1 · 2017-11-27
**Identifies and explores an important issue but lacks in quantitative analysis**

**Rating:** 4
**Confidence:** 3

**Review:**

This paper concerns a potentially serious issue with current GAN based approaches. Complex data distributions, such as natural images, likely lie upon many disconnected manifolds. However standard GANs use continuous noise and generators and must therefore output a connected distribution over inputs. This constraint results in the generator outputting what the paper terms “tunnels” regions of output which connect these actually disconnected manifolds but do not correspond to actual samples from valid manifolds.

This is an important observation. The paper makes a variety of sensible claims - attributing incoherent samples to these tunnels and stating that complex datasets such as Imagenet are more likely to suffer from this problem. This behavior can indeed be observed during training on toy examples such as a 2d mixture of gaussians. However it is an open question how important this issue is in practice and the paper does not clearly separate this issue from the issue of properly modeling the complicated manifolds themselves. It is admittedly difficult to perform quantitative evaluations on generative models but much more work could be done to demonstrate and characterize the problem in practice.

The tunnel problem motivates the authors proposed approach to introducing discontinuities into the generator. Specifically the paper proposes training N different generators composed of N different linear projections of the noise distribution while sharing all further layers. A projection is chosen uniformly at random during training/sampling. An additional extension adds a loss term for the discriminator/generator to encourage predictability and thus diversity of the projection layers and improves results significantly.

The only experimental results presented are qualitative analysis of samples by the authors. This is a very weak form of evidence suffering from bias as the evaluations are not performed blinded and are of a subjective nature. If the paper intends to present experimental results solely on sample quality then, blinded and aggregated human judgments should be expected. As a reader, I do agree that qualitatively the proposed approach produces higher quality samples than the baseline on CelebRoom but I struggle to see any significant difference on celebA itself. I am uncomfortable with this state of affairs and feel the claims of improvements on this task are unsubstantiated.

While discussion is motivated by known difficulties of GANs on highly varied datasets such as Imagenet, experiments are conducted on both MNIST and celebA datasets which are already well handled by current GANs. The proposed CelebRoom dataset (a 50/50 mixture of celebA and LSUN bedrooms) is a good dataset to validate the problem on but it is disappointing that the authors do not actually scale their method to their motivating example. Additionally, utilizing Imagenet would lend itself well to a more quantitative measure of sample quality such as inception score.

On the flip side, a toy experiment with known disconnected manifolds, while admittedly toy could increase confidence since it lends itself to more thorough quantitative analysis. For instance, a mixture of disconnected 2d gaussians where samples can be measured to be on or off manifold could be included.

At a high level I am not as sure as the authors on the nature of disconnected manifolds and the issue of tunnels. Any natural image has a large variety of transformations that can be applied to it that still correspond to valid natural images. Lighting transformations such as brightening or darkening of the image corresponds to a valid image transformations which allows for a “lighting tunnel” to connect all supposedly disjoint image manifolds through very dark/bright images. While this is definitely not the optimal way to approach the problem it is meant as a comment on the non-intuitive and poorly characterized properties of complex high dimensional data manifolds.

The motivating observation is an important one and the proposed solution appears to be a reasonable avenue to tackle the problem. However the paper lacks quantitative evidence for both the importance of the problem and demonstrating the proposed solution.

---

> ### Author Response · Authors · 2017-12-31
> **Reply: Identifies and explores an important issue but lacks in quantitative analysis**
>
> We thank the reviewer for their insightful critique and detailed comments.
> We have added a revision of the paper with additional experiments, minor corrections & clarifications. We realize that there was an error in our discussion concerning unrealistic outputs in DCGANs, and we have withdrawn that section from the paper. However, we would like to point out that this does not detract our main message because this particular proof was meant to mathematically elucidate the problem of tunneling in DCGANs as an example. While our attempt to showcase the problem particularly for DCGANs stands invalidated, the rest of the general arguments set forth in the paper still hold.
> We address other pending concerns below:
>
> Q: While discussion is motivated by known difficulties of GANs on highly varied datasets such as Imagenet, experiments are conducted on both MNIST and celebA datasets which are already well handled by current GANs. The proposed CelebRoom dataset (a 50/50 mixture of celebA and LSUN bedrooms) is a good dataset to validate the problem on but it is disappointing that the authors do not actually scale their method to their motivating example.
>
> A: We have extended our experiments to include results on STL-10, which is a subset of ImageNet. We believe that this is a step towards ImageNet level complexity.
>
> Q: On the flip side, a toy experiment with known disconnected manifolds, while admittedly toy could increase confidence since it lends itself to more thorough quantitative analysis. For instance, a mixture of disconnected 2d gaussians where samples can be measured to be on or off manifold could be included.
>
> A: We thank the reviewer for suggesting the idea of a toy experiment. We have extended our experiments to include results on a toy dataset with 8 equiprobable, concentrated Gaussian distributions. The setup is the same as in the WGAN-GP paper - 8 bivariate Gaussians with means arranged uniformly on a circle of radius 2. The covariance matrices are taken to be 0.02I. We show that our method quickly converges and covers the Gaussians, while standard GAN and WGAN-GP are unable to cover the distribution or take a long time to converge.
>
> Q: At a high level I am not as sure as the authors on the nature of disconnected manifolds and the issue of tunnels. Any natural image has a large variety of transformations that can be applied to it that still correspond to valid natural images. Lighting transformations such as brightening or darkening of the image corresponds to a valid image transformations which allows for a “lighting tunnel” to connect all supposedly disjoint image manifolds through very dark/bright images. While this is definitely not the optimal way to approach the problem it is meant as a comment on the non-intuitive and poorly characterized properties of complex high dimensional data manifolds.
>
> A: The example provided by the reviewer does connect two supposedly disjoint manifolds through very dark images. However, we would like to point out that such images will probably not be part of the "real distribution" of images (of faces, say). Hence it is probably not a real concern that the manifolds will intersect.

---

### Decision · Program_Chairs · 2018-01-29
**ICLR 2018 Conference Acceptance Decision**

**Decision:**

Invite to Workshop Track

**Comment:**

The paper presents a really interesting take on the mode collapse problem and argue that the issue arises because of the current GAN models try to model distributions with disconnected support using continuous noise and generators. The authors try to fix this issue by training multiple generators with shared parameters except for the last layer.

The paper is well written and authors did a good job in addressing some of the reviewer concerns and improving the paper.

Even though arguments presented are novel and interesting, reviewers agree that the paper lacks sufficient theoretical or experimental analysis to substantiate the claims/arguments made in the paper. Limited quantitative and subjective results are not always in favor of the proposed algorithm. More controlled toy experiments and results on larger datasets are needed. The central argument about "tunneling" is interesting and needs deeper investigation. Overall the committee recommends this paper for workshop.